# Finding the Minimal Parameter Budget for Implicit Reasoning: A Data Complexity Driven Scaling Law for Language Models

**Xinyi Wang** [1]  **Shawn Tan** [2]  **Shenbo Xu** [3]  **Mingyu Jin** [4]  **William Yang Wang** [5]  **Rameswar Panda** [2]  **Yikang Shen** [2]

## Abstract

Reasoning is a core capability of language models (LMs), yet it remains unclear how much model capacity is necessary to support reasoning during pretraining. In this work, we study the minimal parameter budget required for implicit reasoning, defined as the ability to infer new facts from learned knowledge without explicit chain-of-thought supervision. To isolate this phenomenon, we pretrain LMs from scratch in a controlled synthetic environment that mimics the structure and distribution of real-world knowledge graphs, and evaluate their ability to complete missing edges via multi-hop inference. From both a theoretical and an empirical perspective, we identify a scaling law linking this optimal parameter budget to a graph search entropy measure. Across a wide range of model sizes, training steps, and graph complexities, we show that an optimally sized language model can reliably reason over approximately 0.008 bits of information per parameter at most. Our results characterize the minimal sufficient capacity for implicit reasoning during pretraining. Our findings provide principled guidance for matching model size to data complexity and offer new insights into the scaling behavior of reasoning in large language models.[1]

## 1. Introduction

Large language models (LMs) exhibit impressive reasoning abilities across domains ranging from mathematics to symbolic manipulation and planning (Wei et al., 2022a;

Guo et al., 2025). While post-training methods such as chain-of-thought prompting and reinforcement learning can substantially enhance reasoning performance (Guo et al., 2025; Yang et al., 2025), these methods operate on top of representations learned during pretraining and at a much smaller scale. This raises a fundamental question: how much model capacity is actually required for reasoning to emerge during pretraining?

The general scaling behavior of LMs at pretraining time has been extensively investigated, including the well-known exponential scaling laws for testing loss and compute proposed by Kaplan et al. (2020) and the training compute-optimal scaling studied by Hoffmann et al. (2022a). Recent work has also examined the scaling of specific capabilities like machine translation (Ghorbani et al., 2022) and knowledge capacity/memorization (Allen-Zhu & Li, 2025; Lu et al., 2024). According to these existing scaling laws, it is in general believed that larger models strictly imply better testing loss or task performance.

In this paper, we aim to find the smallest LM size that can reach the optimal implicit reasoning performance. Following the setup in Wang et al. (2024b), we use **implicit reasoning** to denote the reasoning behavior that naturally emerges during pretraining. i.e. *the capability to draw new conclusions from existing knowledge without being explicitly trained to generate chain-of-thoughts (CoTs).* More specifically, we define implicit reasoning over world knowledge as the task of completing missing edges in an incomplete knowledge graph, which requires multi-hop traversal according to predefined logic rules that are implicitly encoded in the graph generation process. To investigate this, we pretrain LMs from scratch using only triples from the incomplete graph and then evaluate their ability to infer the missing connections.

Our empirical results reveal a striking phenomenon. When training is sufficiently long, the best achievable reasoning performance is determined solely by the data, while the smallest model that achieves this performance—the optimal model size—is finite and stable. Larger models can match this performance but are not required to do so and may overfit under prolonged training. This leads to a characteristic U-shaped test-loss curve as a function of model size.

[1]Princeton Language and Intelligence Lab [2]MIT-IBM Watson AI Lab [3]MIT [4]Rutgers University [5]UC Santa Barbara. Correspondence to: Xinyi Wang <wangxinyilinda@gmail.com>, Yikang Shen <yikang.shn@gmail.com>.

*Proceedings of the 43$^{rd}$ International Conference on Machine Learning*, Seoul, South Korea. PMLR 306, 2026. Copyright 2026 by the author(s).

We formalize these observations in two theoretical results. First, we prove that under a mild separation condition, the budgeted optimal model size converges to a unique global optimal size as training step increases (Theorem 3). This establishes that the notion of a minimal sufficient model for reasoning is well defined and identifiable given enough training. Conceptually, this result connects implicit reasoning to benign overfitting (Bartlett et al., 2020) and double-descent–style phenomena (Nakkiran et al., 2020; Belkin et al., 2020), showing that optimal reasoning emerges at the smallest model capable of representing the task rather than the largest available model.

Second, we address what determines this optimal size. We introduce graph search entropy, an information-theoretic measure that quantifies the uncertainty encountered when traversing a knowledge graph according to its latent rules. We prove that the optimal model size scales linearly with this entropy (Theorem 4), yielding a concrete prediction: the minimal parameter budget required for reasoning is proportional to the intrinsic search complexity of the underlying knowledge graph.

Extensive experiments on both synthetic and real-world knowledge graphs validate these theoretical predictions. In particular, we find a tight empirical scaling law indicating that each parameter in an optimally sized model supports at most $\approx 0.008$ bits of reasoning-relevant information, a quantity dramatically smaller than known memorization capacities.

Together, these results reposition reasoning as a capacity-limited phenomenon governed by data complexity rather than sheer scale, and provide a principled framework for predicting when—and how—smaller models can reason as well as larger ones.

## 2. Problem and Experimental Setup

While real-world LLMs are pretrained on large scale text corpora, this corpus can be viewed as encoding a wide range of world knowledge. The power of LLMs lies in the fact that they can not only memorize the world knowledge and extract the knowledge when queried, but also reason over the world knowledge and draw novel conclusions. In this paper, we propose constructing a simplified pretraining corpus from a knowledge graph. A knowledge graph is comprised of a set of (head entity, relation, tail entity) triples, and we use each knowledge triple as a training example. We test the reasoning capability of a language model trained on such a corpus by testing its accuracy in completing triples that have never been seen in the knowledge graph but can be deduced through latent rules encoded in the graph structure. For example, if we know A is B's father, and B is C's father, then we can deduce that A is C's grandfather.

Formally, a knowledge graph $G$ consists of $|G| = N$ triples $(e^h, r, e^t)$, where $e^h \in \mathcal{E}$ is the head entity, $e^t \in \mathcal{E}$ is the tail entity, and $r \in \mathcal{R}$ is a relation. A simple example of knowledge triple is (DC, is the capital of, USA). These knowledge triples naturally form a graph, with nodes as the entities and each edge labeled with a relation type. We denote the total number of entities or nodes by $|\mathcal{E}| = N_e$ and the total number of edge or relation types by $|\mathcal{R}| = N_r$. Then a corpus constructed from this knowledge graph would consist of $N$ data points. The objective of a language model with with parameter $\theta$ trained on this corpus is then:

$$L(\theta) = \arg \min_{\theta} \frac{1}{N} \sum_{i=1}^{N} - \log P_{\theta}(e_i^h, r_i, e_i^t).$$

Note that $L(\theta)$ is just the next token prediction loss with each triple as a training example. How the loss is computed exactly depends on how the triples are represented and tokenized.

We have tried several ways of representing and tokenizing the triples, including (1) representing each triple as a natural language senetence and tokenize the sentence with a pretrained text tokenizer, and (2) assigning a random integer ID to each entity and relation and concatenating the three ID with a space in between, and then tokenize by character.

As shown in the next section, we find that the second way produce a much cleaner and clearer trend in scaling that enables us to perform more in-depth and regorous analysis. This is likely because the random IDs eliminate confounding variables and information contained in the lexical form of the entity and relation names,

We use the Llama (Touvron et al., 2023) model architecture to implement LMs of different sizes by adjusting the hidden dimensions and the number of layers. The specific parameter scheme can be found in the Section E.

To evaluate the language model's capability of reasoning over the knowledge graph, we test the LMs on a held-out set of triples that are not seen in the training time. Note that all entity and relation types should have been seen during training time and the language model is only tasked to connect missing edges. To eliminate the need to generate the correct form of relation and entity IDs, we design the testing set to be 10-option multiple-choice questions: the language model is tasked to choose the correct tail entity given the head entity and the relation. We ensure that there is only one correct answer among the given 10 options. Suppose there are $M$ questions in the testing set.[2] For a ground truth triple $(e^h, r, e^t)$, we randomly sample 9 distracting options $e^{(1)}, e^{(2)}, ..., e^{(9)}$ that are not the correct answer. Then we use the test accuracy $\text{Acc}(\theta, G)$ and testing loss $\ell(\theta, G)$ (the next token prediction loss) to evaluate the reasoning

---

[2]We fix $M = 1000$ for all of our experiments.

capability of a language model $\theta$ over the knowledge graph $G$:

$$\hat{e}_i = \arg \max_{e \in \{e_i^t, e_i^{(1)}, e_i^{(2)}, \ldots, e_i^{(9)}\}} P_\theta(e|e_i^h, r_i),$$

$$\mathrm{Acc}(\theta, G) = \sum_{i=1}^{M} \mathbb{1}[\hat{e}_i = e_i^t]/M,$$

$$\ell(\theta, G) = \sum_{i=1}^{M} - \log P_\theta(e_i^t|e_i^h, r_i)/M.$$

## 3. Optimal Model Size Analysis

In our initial sets of experiments, we investigate the reasoning scaling effect using a real-world knowledge graph, FB15K-237 (Toutanova & Chen, 2015). FB15K-237 is sampled from FB15K (Bordes et al., 2013), which is a dataset adapted from the Freebase knowledge base (Bollacker et al., 2007), a web-scale knowledge base released by Google. FB15K-237 contains $N_e = 14,505$ entities, $N_r = 237$ relations, and $N = 310,116$ knowledge triples. We process this dataset in three different ways: (a) translate each knowledge triple into a natural language sentence by prompting GPT4 and then tokenize the sentence with a pre-trained tokenizer, as shown in the first row of Figure 1; (b) translate each knowledge triple into a natural language sentence using pre-generated templates, as show in the second row of Figure 1; (c) translate each knowledge triple into text by assigning a random ID to each entity and relation and tokenize them by characters, as shown in the last row of Figure 1. Data processing examples can be found in Section D Figure 5.

In Figure 1, we show different-sized LMs trained on FB15K-237 in all settings with different numbers of training steps. For all experiments, we use a constant batch size 1024, which we found produce optimal performance. So the number of training steps is directly comparable across different experiments. While the training loss decreases monotonically with respect to model size, we observe that the implicit reasoning testing loss does not monotonically decrease across different settings, especially when the number of training steps is large. We also observe that the maximum implicit reasoning performance is not necessarily reached by the largest model.

While the trends of the first two rows with neutral language settings are somewhat noisy, the last row with random IDs shows a clear and reproducible pattern: across runs, the *lowest achievable* test loss (and the highest achievable reasoning accuracy) is largely stable, whereas the model size that attains it may vary. Moreover, as the training budget increases, the smallest model size that can achieve near-optimal test performance stabilizes. This motivates defining an **optimal model size** based on the best checkpoint (early stopping) rather than the loss at a fixed training step, which

may later deteriorate due to overfitting.

As $\theta$ denotes the LM parameters and $\theta_t$ denotes the parameters after $t$ training steps, let $N_\theta$ be the number of parameters in the model and $\ell(\theta_t, G)$ be the test loss on knowledge graph $G$. Here we first formally define the **best-achievable test loss** and the **optimal model size**:

**Definition 1** (Best-achievable test loss). *For any model parameters $\theta$, define the best-achieved test loss up to step $t$ as*

$$\underline{\ell}_t(\theta, G) := \min_{0 \leq s \leq t} \ell(\theta_s, G),$$

*and the best-achieved test loss over an unbounded training horizon as*

$$\underline{\ell}_\infty(\theta, G) := \inf_{s \geq 0} \ell(\theta_s, G).$$

*Define the globally optimal achievable test loss as*

$$\underline{\ell}_\infty^*(G) := \inf_\theta \underline{\ell}_\infty(\theta, G).$$

**Definition 2** (Optimal model size). [3] *Fix $\epsilon > 0$. The (global) $\epsilon$-optimal model size for $G$ is defined as*

$$N_\theta^*(G) := \min \left\{ N_\theta : \exists \theta, \; \underline{\ell}_\infty(\theta, G) \leq \underline{\ell}_\infty^*(G) + \epsilon \right\}.$$

*For a finite training budget $t$, define the budgeted optimal achievable loss*

$$\underline{\ell}_t^*(G) := \inf_\theta \underline{\ell}_t(\theta, G),$$

*and the budgeted $\epsilon$-optimal model size*

$$N_{\theta,t}^*(G) := \min \left\{ N_\theta : \exists \theta, \; \underline{\ell}_t(\theta, G) \leq \underline{\ell}_t^*(G) + \epsilon \right\}.$$

Then we have the following theorem stating that the budgeted optimal model size at $t$-th step approaches the global optimal model size as $t$ grows, under a mild gap condition in best-achieved test loss. This gap condition can be understood as a result of the U-shaped loss curve.

**Theorem 3** (Convergence of budgeted optimal model size). *Fix a knowledge graph $G$ and $\epsilon > 0$. Assume there exists $\Delta > 0$ such that for the global optimal size $N_\theta^*(G)$, every strictly smaller model size $n < N_\theta^*(G)$ satisfies*

$$\inf_{\theta:N_\theta = n} \underline{\ell}_\infty(\theta, G) \; \geq \; \underline{\ell}_\infty^*(G) + \epsilon + \Delta.$$

*Then $N_{\theta,t}^*(G)$ converges and*

$$\lim_{t \to \infty} N_{\theta,t}^*(G) = N_\theta^*(G).$$

---

[3]The tolerance $\epsilon$ accounts for finite-sample evaluation noise and flat generalization plateaus. If the globally optimal test loss is attained uniquely by a smallest model size and is separated from all smaller models by a strict margin, the results hold with $\epsilon = 0$.

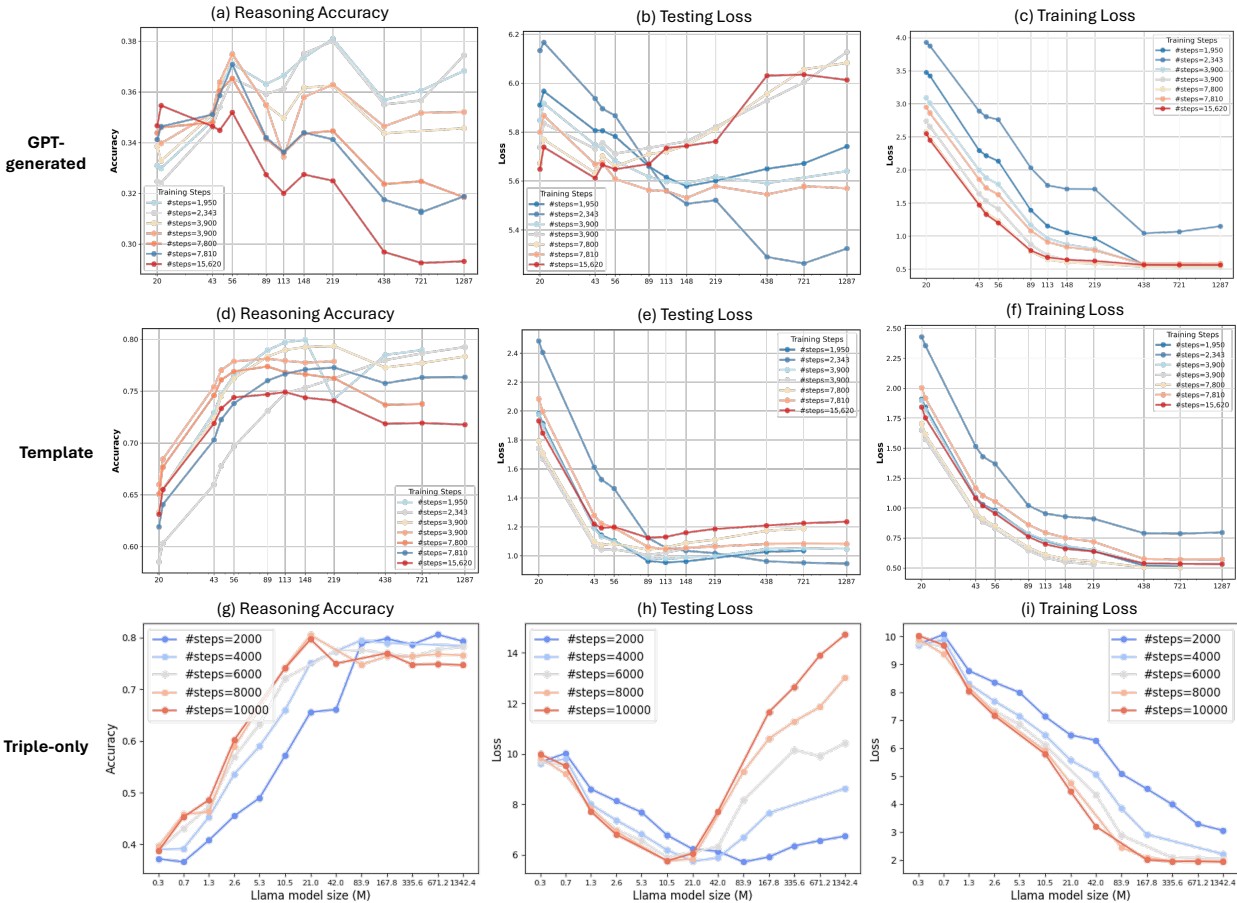

*Figure 1.* The multiple-choice accuracy/loss on unseen triples of different-sized LMs trained on a real-word knowledge graph FB15K-237. The first column shows that the testing accuracy decreases after a certain model size. The second column shows U-shape loss curves of LMs trained with different numbers of steps. The third column shows the training loss decreases steadily. These trends are stable across different ways of processing the knowledge triples, with the triple-only data shows the cleanest trend. Note that the model size on x-axis is in log scale.

Theorem 3 shows that larger models can match optimal performance – they are simply not required. The U-shaped curve arises only under prolonged training where overfitting occurs.

In the following sections, we focus on the random-ID setting, where the best-achieved test performance is stable across runs and the induced optimal model size is empirically well defined.

Similar unconventional scaling laws have also been reported in broken neural scaling law (Caballero et al., 2023) which proposed a double-descent (Nakkiran et al., 2020; Belkin et al., 2020) function form instead of a monotonic power-law form. There have also been observations of tasks with inverse scaling (Wei et al., 2023) for large LMs. Our U-shaped scaling curve here is a result that we push the training condition to the extreme to test the limit of small LMs. For larger LMs, smaller number of training steps can yield much better performance as suggested in Figure 1. Our theorem of optimal model size formalizes the notion of a minimal benignly

overfitting model (Bartlett et al., 2020): Larger models may achieve comparable test loss but are not required to do so, and may overfit if trained excessively. Our convergence result shows that, under random ID tokenization, this minimal benignly overfitting model size is well defined and can be identified with sufficient training budget.

In this paper, we focus primarily on the scaling of model size and data complexity. Rather than merely increasing the size of the training data, we explore different schemes for generating synthetic knowledge graphs. This allows us to ablate individual components of the generation process and examine how graph complexity affects reasoning. Our goal is to predict the optimal model size for implicit reasoning given a knowledge graph.

## 4. Synthetic Data Construction

To investigate how the underlying knowledge structure influences LMs' reasoning performance, we propose an algorithm to generate synthetic knowledge graphs that mimic

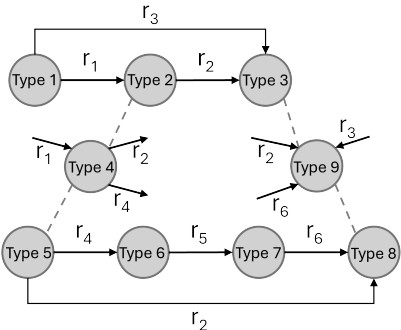

*Figure 2.* Nine possible node types generated by two logical rules. Each entity position in a rule would create a new entity type. Each relation shared between two rules would also create two new entity types.

real-world knowledge graphs. More specifically, we assume that the knowledge graph generation process is governed by a set of logical rules.

For example, a rule for inferring the `locatedIn` relation can be $(e_1, \texttt{locatedIn}, e_2) \leftarrow (e_1, \texttt{neighborOf}, e_3) \wedge (e_3, \texttt{locatedIn}, e_2)$. Formally, for a target relation $r$, we consider logic rules with conjunctive form:

$$\forall \{e_i\}_{i=0}^n \subset \mathcal{E}, (e_0, r, e_n) \leftarrow (e_0, r_1, e_1) \wedge ... \wedge (e_{n-1}, r_n, e_n),$$

where $(e_{i-1}, r_i, e_i) \in \mathcal{G}$. We abbreviate such rule by $h(r) = [r_1, r_2, ..., r_n]$. We randomly generate a set of logical rules $\mathcal{H}$ and ensure there are no cycles in the set. To grow a graph that follows these rules, we enforce sparsity of the possible relation types connecting to and branching out each entity. More specifically, we define *node types* based on the possible relation types connecting to and branching out each entity, based on the generated rules, as illustrated in Figure 2. Such sparsity is also observed in real-world knowledge graphs.

Our random graph generation process is inspired by the preferential attachment process (Barabási & Albert, 1999), which is used for generating scale-free networks with a power-law distribution for the degrees of the nodes. Intuitively, preferential attachment implies a "the rich get richer" approach to edge placement in the graph. Each time a new node is added to the graph, there is a 'preference' to connect to the nodes that are already highly connected, with a probability proportional to the target node's degree. Since we have observed the scale-free property in real-world knowledge graphs and the internet is known to be a scale-free network (Albert et al., 1999), we adopt a preferential attachment based graph generation process. To accommodate different relation types assigned to each edge, we maintain a degree distribution for each relationship and add new edges according to preferential attachment. A comparison of the node degree distribution between synthetic graph and real-world

graph can be found in Section F Figure 6.

The code for our random graph generation algorithm is shown in the Section G. In summary, we first randomly generate a set of rules $\mathcal{H}$, with the number of rules $|\mathcal{H}| = N_h$ and the range of rule length $[L_{min}, L_{max}]$ as hyperparameters. Then we generate all possible node types as illustrated in Figure 2, with the maximum number of relations per node $M_r$ as a hyperparameter. We generate a seed graph by instantiating each rule with a set of new entities, and then incrementally add one new entity until the number of entities reaches $N_r$, by first randomly assigning a node type to it, and then randomly sampling the $m$ relation types from the set of relations defined by the node type. We choose the target of these $m$ new edges by preferential attachment. After adding every $K$ entities, we search the current graph to add any edges that can be inferred with logic rules defined in $\mathcal{H}$. We call the triples that can be deduced through a logic rule by *deducible triples*, otherwise *atomic triples*.

Finally, we limit the number of training triples to $N$ and ensure that the the ratio between the number of deducible triples and atomic triples to $\gamma$ by subsampling the generated graph. We also further ensure that the triples in the held-out test set are all deducible through the training triple. In this way, we can generate synthetic knowledge graphs with specific sizes and complexity.

## 5. Scaling Laws

In this section, we investigate the scaling law of language models trained on different synthetic knowledge graphs. We conduct controlled experiments to show the effect of individual components of the data generation process. We also propose an information-theoretical way to measure the overall reasoning complexity of a knowledge graph, which we call the **graph search entropy**, and relate this linearly with the **optimal model size**. i.e. the model size that obtains the lowest possible testing loss.

### 5.1. Graph Generation Ablation

We study how the optimal model size varies as we independently modify key properties of the synthetic knowledge graphs. We fix all training hyperparameters as specified in the Section E. Except for Figure 3(a), we train for 10k steps as a practical approximation to the large-budget regime implied by Theorem 3. The detailed data generation configuration for each set of experiments can also be found in the Section E.

- **Training steps** $t$ **(a):** Increasing training steps reduces the smallest model size that attains the best achievable reasoning loss, after which the optimal size stabilizes (Figure 3a). This confirms that the optimal model size

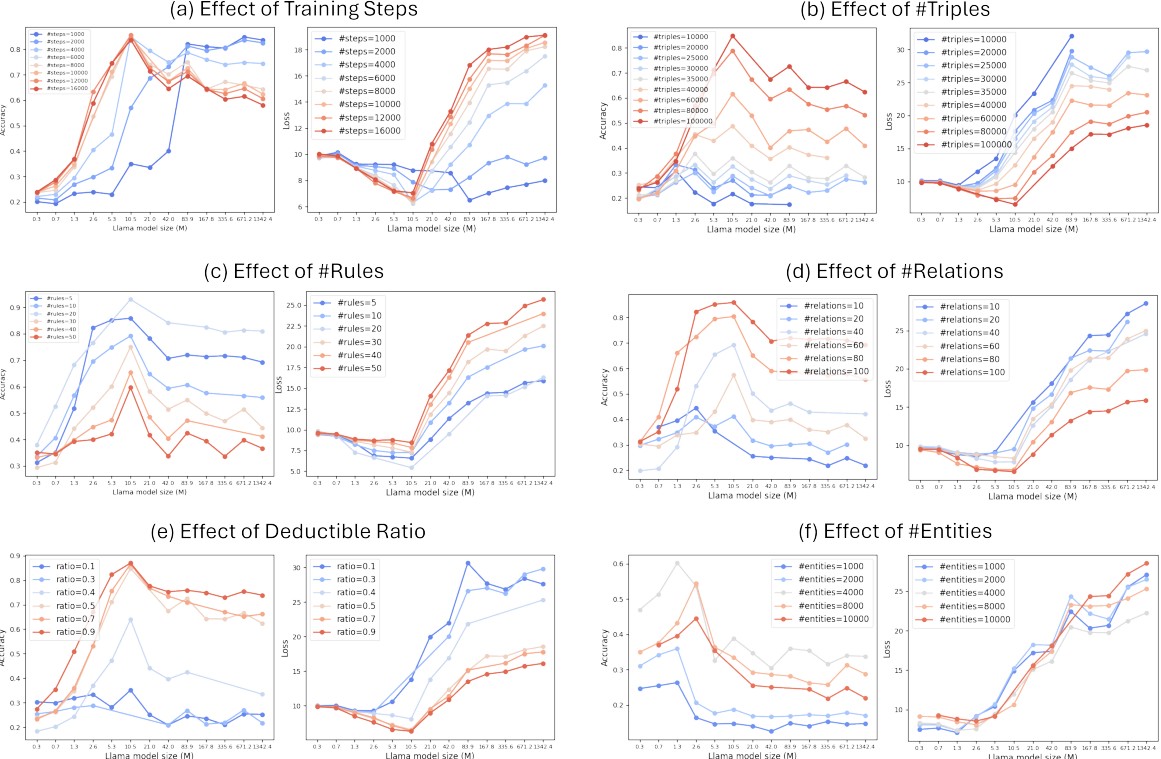

*Figure 3.* We show the effect of different hyperparameters of the synthetic knowledge graph generation process. In each experiment, we keep all other parameters the same and only change one hyperparameter. We show the effect with both the testing accuracy (left) and the testing loss (right) as the y-axis, with different model sizes as the x-axis in log scale.

is a property of the data rather than the training budget, consistent with Theorem 3.

- **Number of triples** $N$ **(b):** Increasing the number of training triples increases both reasoning performance and the optimal model size (Figure 3b). This mirrors classical scaling behavior and reflects the increased information content exposed during training.

- **Number of rules** $N_h$ **(c):** Varying the number of logical rules has little effect on the optimal model size, though it impacts reasoning accuracy (Figure 3c). This suggests that rule multiplicity affects ambiguity and solvability but does not substantially change the underlying search complexity of the graph.

- **Number of relations** $N_r$ **(d):** Graphs with more relation types require larger optimal models and achieve better reasoning performance (Figure 3d). Additional relations increase graph complexity while reducing spurious correlations, making rule-based reasoning more identifiable.

- **Deducible ratio** $\gamma$ **(e):** Increasing the ratio of deductible to atomic triples improves reasoning performance and increases optimal model size when the ratio is small, after which both effects saturate (Figure 3e).

Once rule patterns are sufficiently exposed, additional deductible triples yield diminishing returns.

- **Number of entities** $N_e$ **(f):** Increasing the number of entities increases the optimal model size while generally degrading reasoning performance when rules and relations are sparse (Figure 3f). Larger graphs increase search complexity but also amplify ambiguity under limited rule structure.

Across all ablations, the optimal model size consistently tracks changes in the effective search complexity of the knowledge graph, motivating a unified complexity measure formalized in the next section.

### 5.2. Optimal Model Size v.s. Graph Search Entropy

From our previous ablation studies, we hypothesize that the optimal model size is positively related to the overall complexity of the knowledge graph. Thus, we propose that we measure the complexity of a knowledge graph by quantifying the amount of information that can be obtained from the graph by exploring the graph through a random search. From our task definition, to reason over the knowledge graph, the language model needs to (a) identify the set of logic rules by observing repetitive patterns; (b) traverse

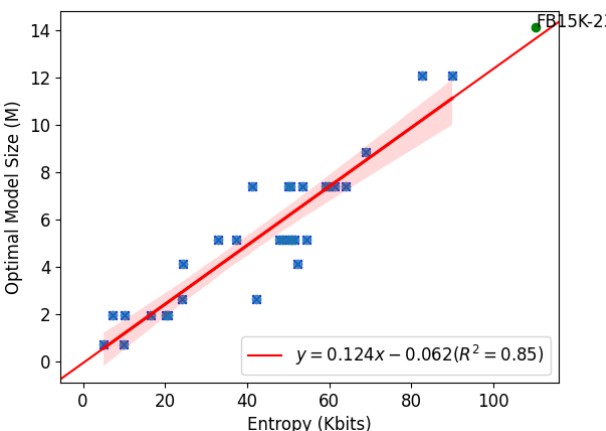

*Figure 4.* The optimal model size with the lowest possible testing loss v.s. the graph search entropy. The red line is the linear regression line using data from the synthetic experiments (blue squares), with a 95% confidence interval. We also plot the graph search entropy and optimal model size from the real-world FB15K-237 experiment (green dot) to verify the accuracy of the obtained linear scaling law.

the graph using one or more specific logic rules to locate the tail entity. So we define the **graph search entropy** as the maximum amount of information that can be obtained when randomly traversing the graph.

To simplify the problem, we first focus on the average amount of information we can observe at one node of the graph. If we consider a random walk over the knowledge graph, then we refer to the entropy produced by each node on the walk trace for an infinitely long random walk as the *entropy rate* of this random walk. For a graph $G$, the maximum entropy rate is equal to the log of the largest eigenvalue of the adjacency matrix $A$. This gives us the entropy rate with respect to the entity, without considering the entropy rate with respect to the relation. We can compute the relation entropy rate with the stationary distribution and transition matrix induced by the maximal entropy rate random walk. If we denote the dominating eigenvalue by $\lambda \in \mathbb{R}$ and the corresponding eigenvector by $\psi \in \mathbb{R}^{N_e}$, then the stationary distribution $\rho \in \mathbb{R}^{N_e}$ can be written as:

$$\rho_i = \psi_i^2/||\psi||_2^2.$$

The transition matrix $S \in \mathbb{R}^{N_e \times N_e}$ of the maximal entropy random walk can be written as:

$$S_{ij} = (A_{ij}/\lambda)(\psi_j/\psi_i).$$

We can then transform the entity-to-entity transition matrix $S \in \mathbb{R}^{N_e \times N_e}$ into an entity-to-relation transition matrix $S^r \in \mathbb{R}^{N_e \times N_r}$ by merging entries with the same relation:

$$S_{ij}^r = \sum_{k=1}^{N_e} \mathbb{1}[(i,j,k) \in G]S_{ik}.$$

Finally, the relation entropy rate $H^r(G)$ can be written as:

$$H^r(G) = -\sum_{i=1}^{N_e} \rho_i \sum_{j=1}^{N_r} S_{ij}^r \log(S_{ij}^r).$$

The overall **graph search entropy** $H(G)$ can then be written as the sum of the entity entropy rate and the relation entropy rate over all entites:

$$H(G) = N_e \cdot (\log \lambda + H^r(G)).$$

To our knowledge, this particular formulation – designed to quantify how much information a model must internalize to perform multi-hop reasoning over a knowledge graph – has not appeared in prior work. Because the total conditional complexity of a graph scales linearly with graph search entropy (full proof see Section C), so we have the following theorem:

**Theorem 4** (Optimal model size scales with graph search entropy). *Let $G$ be a knowledge graph. Let $p_G(\cdot \mid x)$ denote the Bayes-optimal conditional distribution over the tail entity $Y$ given the input entity $X = x$, and define the total conditional complexity*

$$C(G) := \sum_{x \in \mathcal{E}} H(Y \mid X = x).$$

*Assume the following conditions hold.*

*(i)* **No semantic sharing across entities.** *Entity identifiers are random IDs, so predictors for different heads cannot be compressed through semantic overlap.*

*(ii)* **Finite-precision parameters.** *An $N$-parameter model has effective information capacity $O(N)$.*

*(iii)* **Sparse basis approximation of conditionals.** *For each entity $x \in \mathcal{E}$, the Bayes conditional $p_G(\cdot \mid x)$ admits an approximation*

$$\tilde{p}_x = \text{softmax}(B^\top a_x)$$

*such that*

$$D_{\text{KL}}(p_G(\cdot \mid x)\|\tilde{p}_x) \le \epsilon_x, \qquad \sum_x \pi(x)\epsilon_x \le \epsilon,$$

*where $B \in \mathbb{R}^{r \times |Y|}$ is shared and*

$$\|a_x\|_0 \le \alpha H(Y \mid X = x) + \beta.$$

*for constants $\alpha, \beta > 0$ independent of $x$.*

*Then the $\epsilon$-optimal model size satisfies*

$$N_\theta^*(G) = \Theta(H(G)).$$

*Remark* 1. Relaxing Assumption (i) would tighten, not loosen, the bound. Assumption (i) establishes an upper bound on the minimal parameter budget: since semantic overlap provides additional signal that can compress entity representations, the true optimal size for natural language data should be at most what our theory predicts.

*Remark* 2. The upper bound is architecture-dependent: Assumption (iii) implies a constructive upper bound for an explicit one-layer Transformer under a stronger sparse-basis condition. This matches our empirical setup, where all models use Transformer architectures. The lower bound is more architecture-agnostic, relying mainly on random IDs and finite-precision parameters, but the matching upper bound depends on the ability of attention to retrieve sparse head-specific coefficients through learned key-value memories. An important direction for future work is to test whether the same scaling law holds for non-attention architectures, especially state-space models (SSMs) (Gu et al., 2022; Gu & Dao, 2024).

We empirically investigate the relation between the optimal model and the graph search entropy by plotting them against each other in Figure 4, and perform linear regression. The optimal model sizes are obtained from the synthetic experiments conducted in the ablation studies. In the ablation studies we only report the results for exponentially increasing model sizes for clarity. In this study to better capture the optimal model size, we make the model sizes near the optimal model size more fine-grain.

We find a strong linear relation between the optimal model size and the graph search entropy with $R^2 = 0.85$. Note that there are a few sources of noise for locating the optimal model size for a specific knowledge graph. First, we only train language model with selected sizes due to compute and time limitations, and the discretization of the model size would disrupt the smoothness of the scaling law. Second, we cannot train for infinitly long time and choose to inspect at the training step 10k.

After fitting a linear regression line using the data from our synthetic experiments, we check the validity of this empirical scaling law against our real-world knowledge graph, FB15K-237. We calculate the graph search entropy for FB15K-237, and find the predicted optimal model size is very close to the observed optimal model size, shown as a green dot in Figure 4.

From our scaling law, we can see that roughly 124 additional parameters in the optimal model size are required per 1-bit entropy increase in the knowledge graph. That is a language model can only reliably (not perfectly) reason over 0.008 bit information per parameter. This is very different from the knowledge capacity scaling law concluded by (Allen-Zhu & Li, 2025), which shows that the language model can store 2 bits of knowledge per parameter. We

think this discrepancy is due to two reasons: first, our scaling law is not only about memorizing the knowledge, but also about reasoning over the learned knowledge, which is significantly harder. Second, the way we compute the graph search entropy is fundamentally different from the way (Allen-Zhu & Li, 2025) computes the knowledge entropy. While (Allen-Zhu & Li, 2025) describes the entropy of the knowledge generation process, our graph search entropy describes the entropy of randomly traversing a fixed knowledge graph. In this way, we did not directly measure the amount of information that a language model needs to memorize, but measuring the complexity of traversing, and therefore, reasoning over a graph. It is hard, if not impossible, to obtain the data generation process of real-world data, but it is possible to get an estimate of the underlying knowledge graph of a corpus through automated knowledge graph construction algorithms (Zhong et al., 2023). Thus, it is possible to predict the optimal reasoning model size for real-world pretraining, by first constructing a knowledge graph from the pretraining corpus, and then computing its graph search entropy, and finally using a similar scaling law to calculate the optimal model size.

## 6. Related Work

**Language Model Scaling Laws** (Kaplan et al., 2020) first observed a power-law relationship between LLM perplexity, model parameter count, and training data size, laying the foundation for scaling law research. Subsequently, (Hoffmann et al., 2022b) explored optimal training strategies under constrained computational resources and discovered that LLM parameter size and the number of training tokens should scale proportionally to achieve optimal compute efficiency under a fixed budget. Beyond pretraining performance, researchers further confirmed that downstream task performance can also be reliably predicted based on model size and training data volume (Hernandez et al., 2021; Isik et al., 2024). (Allen-Zhu & Li, 2025; Lu et al., 2024) have turned to exploring more specific capability dimensions, focusing particularly on the scaling laws of factual memory in LLMs and their behavioral patterns when memorizing different types of facts. Most recently, (Roberts et al., 2025) have confirmed that scaling laws are skill-dependent, and found that knowledge-intensive tasks are more parameter-hungry while reasoning-intensive tasks are more data-hungry. (Springer et al., 2025) challenge a core assumption in scaling research—that more pretraining invariably leads to better downstream performance. Our paper identifies a different U-shaped scaling curve under the specific scenario of knowledge graph reasoning and reveals that the search complexity of the knowledge graph determines the optimal model size. This echoes the discovery of (Pandey, 2024) and (Yin et al., 2024) that classic scaling laws are highly dependent on the data complexity or the

compression ratio of the data. (Havrilla & Liao, 2024) also confirmed from both theoretical and empirical perspectives that the power of the power scaling law depends on the intrinsic dimension of the training data.

**Language Model Reasoning**  Our paper focuses on the reasoning capability of LMs which has drawn a lot of attention recently (Zhang et al., 2023; Chen et al., 2023; Yao et al., 2023a;b; Wang et al., 2023; Guo et al., 2025; Jin et al., 2024; Yeo et al., 2025; Team et al., 2025; Li et al., 2025). LLMs are usually trained to reason in a step-by-step manner in real-world tasks like math problems (Wei et al., 2022b) and coding (Yang et al., 2024). In our experiments, we do not ask LMs to generate a CoT solution, but ask the language model to directly choose the correct answer from the given options, because our pretrain-only LMs are not trained to give a CoT solution for a query. Our synthetic reasoning environment is the most similar to (Wang et al., 2024b), which also use the knowledge graph completion task as a testbed to understand how LMs learn to reason at pretraining time. They propose that LMs are able to aggregate random walk paths sampled from the knowledge graph. (Wang et al., 2024a; Zhu et al., 2024) also employ a graph structure to ground their synthetic reasoning tasks to explain how LLMs reason, but their reasoning is defined as concatenations of relations: A is $r_1$ to B and B is $r_2$ to C implies A is $r_1 r_2$ to C. The knowledge graph completion task we employ is more complex than simple concatenation of relations as the language model needs to find out which relation $r_1 r_2$ corresponds to from the knowledge graph.

## 7. Conclusion

We studied how implicit reasoning scales during pretraining by training language models from scratch on knowledge-graph triples and evaluating unseen triples prediction. Empirically, we find a non-monotonic scaling pattern: given sufficient optimization, smaller models can reach the same minimum reasoning loss as larger models on a fixed graph, suggesting a well-defined *optimal* (smallest sufficient) model size. We then connect this optimal size to data complexity via graph search entropy, and show theoretically and empirically an approximately linear scaling law between graph search entropy and the optimal parameter budget. In our setting, this relationship implies a reasoning capacity of at most $\approx 0.008$ bits of graph search information per parameter.

## Impact Statement

This work advances understanding of how reasoning capabilities emerge during language model pretraining by revealing a non-monotonic relationship between model scale and implicit reasoning performance. By identifying an optimal

model size tied to data complexity rather than sheer scale, our findings may encourage more compute-efficient and environmentally sustainable model design, helping reduce unnecessary resource consumption and lowering barriers to participation in large-scale AI research.

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

# Appendix

## A. Limitations

We want to highlight that this study is only conducted on simplified pretraining data from knowledge graphs, and the results are not directly applicable to real-world language model pretraining with large text corpus. The setting of our study provides a reasonable analogy to the real-world language model pretraining, and the obtained insight might be found useful in the real world when the compute is abundant with very large models and very large datasets that exhaustively traverse the underlying knowledge graph. We leave the work of verifying our scaling law in the real word to future research due to its resource-demanding nature.

## B. Proof of Theorem 3

**Theorem 5** (Convergence of budgeted optimal model size). *Fix a knowledge graph $G$ and $\epsilon > 0$. Assume there exists $\Delta > 0$ such that for the global optimal size $N_\theta^*(G)$, every strictly smaller model size $n < N_\theta^*(G)$ satisfies*

$$\inf_{\theta:N_\theta=n} \ell_\infty(\theta, G) \geq \ell_\infty^*(G) + \epsilon + \Delta.$$

*Then $N_{\theta,t}^*(G)$ converges and*

$$\lim_{t \to \infty} N_{\theta,t}^*(G) = N_\theta^*(G).$$

*Proof.* First, note that $\underline{\ell}_t(\theta, G)$ is non-increasing in $t$ for any fixed $\theta$, hence $\underline{\ell}_t(\theta, G) \downarrow \underline{\ell}_\infty(\theta, G)$ as $t \to \infty$. Taking infima over $\theta$ yields $\underline{\ell}_t^*(G) \downarrow \underline{\ell}_\infty^*(G)$.

Let $n^* := N_\theta^*(G)$. By definition of $n^*$, there exists a parameterization $\theta^*$ with $N_{\theta^*} = n^*$ such that for any $\delta > 0$, there exists a finite step $t^*$ satisfying

$$\underline{\ell}_{t^*}(\theta^*, G) \leq \underline{\ell}_\infty(\theta^*, G) + \delta \leq \underline{\ell}_\infty^*(G) + \epsilon + \delta.$$

We can simply absorb $\delta$ into $\epsilon$. Hence for all $t \geq t^*$,

$$\underline{\ell}_t(\theta^*, G) \leq \underline{\ell}_\infty^*(G) + \epsilon.$$

Since $\underline{\ell}_t^*(G) \geq \underline{\ell}_\infty^*(G)$, we obtain

$$\underline{\ell}_t(\theta^*, G) \leq \underline{\ell}_t^*(G) + \epsilon,$$

so $N_{\theta,t}^*(G) \leq n^*$ for all $t \geq t^*$.

On the other hand, for any size $n < n^*$ and any $\theta$ with $N_\theta = n$, we have $\underline{\ell}_t(\theta, G) \geq \underline{\ell}_\infty(\theta, G)$, thus by the gap assumption,

$$\underline{\ell}_t(\theta, G) \geq \underline{\ell}_\infty^*(G) + \epsilon + \Delta.$$

Since $\underline{\ell}_t^*(G) \downarrow \underline{\ell}_\infty^*(G)$, there exists $T$ such that for all $t \geq T$, $\underline{\ell}_t^*(G) \leq \underline{\ell}_\infty^*(G) + \Delta/2$. Therefore, for all $t \geq T$ and all sizes $n < n^*$,

$$\inf_{\theta:N_\theta=n} \underline{\ell}_t(\theta, G) \geq \underline{\ell}_t^*(G) + \epsilon + \Delta/2,$$

so no model with fewer than $n^*$ parameters can be $\epsilon$-optimal at budget $t \geq T$. Hence $N_{\theta,t}^*(G) \geq n^*$ for all $t \geq \max\{T, t^*\}$.

Combining both directions shows that for all sufficiently large $t$, $N_{\theta,t}^*(G) = n^*$, which implies $\lim_{t\to\infty} N_{\theta,t}^*(G) = N_\theta^*(G)$. $\qquad\square$

## C. Proof of Theorem 4

**Theorem 6** (Optimal model size scales with graph search entropy). *Let $G$ be a knowledge graph. Let $p_G(\cdot \mid x)$ denote the Bayes-optimal conditional distribution over the tail entity $Y$ given the input entity $X = x$, and define the total conditional complexity*

$$C(G) := \sum_{x \in \mathcal{E}} H(Y \mid X = x).$$

*Assume the following conditions hold.*

(i) **No semantic sharing across entities.** *Entity identifiers are random IDs, so predictors for different heads cannot be compressed through semantic overlap.*

(ii) **Finite-precision parameters.** *An $N$-parameter model has effective information capacity $O(N)$.*

(iii) **Sparse basis approximation of conditionals.** *For each entity $x \in \mathcal{E}$, the Bayes conditional $p_G(\cdot \mid x)$ admits an approximation*

$$\tilde{p}_x = \mathrm{softmax}(B^\top a_x)$$

*such that*

$$D_{\mathrm{KL}}(p_G(\cdot \mid x)\|\tilde{p}_x) \leq \epsilon_x, \qquad \sum_x \pi(x)\epsilon_x \leq \epsilon,$$

*where $B \in \mathbb{R}^{r \times |Y|}$ is shared and*

$$\|a_x\|_0 \leq \alpha H(Y \mid X = x) + \beta.$$

*for constants $\alpha, \beta > 0$ independent of $x$.*

*Then the $\epsilon$-optimal model size satisfies*

$$N_\theta^*(G) = \Theta(H(G)).$$

We provide a full proof of Theorem 4.

Let $G = (\mathcal{E}, \mathcal{R}, \mathcal{G})$ be a synthetic knowledge graph with $|\mathcal{E}| = N_e$. A labeled example is generated by sampling a head entity $X = E_0$ (random query distribution), sampling a rule length $L \sim \mathcal{D}_L$ supported on $[L_{\min}, L_{\max}]$, and sampling a latent rule instantiation that yields a tail entity $Y = E_L$. The rule chain is *not* provided as part of the input. Let $p_G(\cdot \mid x)$ denote the induced conditional distribution of $Y$ given $X = x$.

Evaluation uses cross-entropy loss. The Bayes-optimal conditional predictor is $p_G(\cdot \mid x)$ and the Bayes risk equals

$$\ell^*(G) = \mathbb{E}_X\big[H(Y \mid X)\big] = \sum_{x \in \mathcal{E}} \pi(x)\, H(Y \mid X = x).$$

Note that $\ell^*(G)$ is an average over heads, while the object governing representation complexity in the random-ID setting is the *sum* over heads:

$$\mathcal{C}(G) := \sum_{x \in \mathcal{E}} H(Y \mid X = x).$$

We will prove that model size scales with the *total conditional complexity*.

## C.1. Model and optimal size

Let $\mathcal{M}_N$ be the model class with $N$ parameters (under a fixed architectural family). Let $\ell(f; G)$ denote the expected test cross-entropy of predictor $f \in \mathcal{M}_N$. Define the $\epsilon$-optimal model size as

$$N_\theta^*(G) := \min\{N : \exists f \in \mathcal{M}_N \text{ s.t. } \ell(f; G) \leq \ell^*(G) + \epsilon\}.$$

## C.2. Assumptions

We use three assumptions, matching the theorem statement.

**(A1) No semantic sharing across heads (random IDs).** Because entities are random IDs, achieving $\epsilon$-excess cross-entropy requires representing the family $\{p_G(\cdot \mid x)\}_{x \in \mathcal{E}}$ with description length $\Omega(\mathcal{C}(G))$.

**(A2) Linear information capacity in parameter count.** An $N$-parameter model class has effective information capacity $O(N)$ (e.g., under finite precision, the number of distinguishable predictors is at most exponential in $N$).

**(A3) Sparse basis approximation of conditionals.** There exists a one-layer attention model with learned key-value memory and shared output matrix $B^\top$ whose expected cross-entropy is at most $\ell^*(G) + \epsilon + \delta$, for arbitrarily small attention error $\delta$, and whose parameter count is

$$O\left(\sum_x \|a_x\|_0 + |\mathcal{E}| + r|Y|\right) = O\left(\mathcal{C}(G) + |\mathcal{E}| + r|Y|\right).$$

**C.3. Lower bound:** $N_\theta^*(G) \geq c_1 \mathcal{C}(G) - O(1)$

**Lemma 7.** *Under (A1)–(A2), there exists $c_1 > 0$ such that*

$$N_\theta^*(G) \geq c_1 \mathcal{C}(G) - O(1).$$

*Proof.* Fix $\epsilon > 0$. Consider the set of conditional distributions $\{p_G(\cdot \mid x)\}_{x \in \mathcal{E}}$. Under (A1), to achieve $\epsilon$-excess cross-entropy, a predictor must encode this family to accuracy $\epsilon$ on typical heads, which requires $\Omega(\mathcal{C}(G))$ bits of task-specific information.

Under (A2), an $N$-parameter model class can encode at most $O(N)$ bits of task-specific information. Therefore any model achieving $\epsilon$-excess cross-entropy must satisfy $O(N) \geq \Omega(\mathcal{C}(G))$, i.e., $N \geq c_1\mathcal{C}(G) - O(1)$ for some constant $c_1 > 0$. By definition of $N_\theta^*(G)$ as the minimal such $N$, the claim follows. $\square$

**C.4. Upper bound:** $N_\theta^*(G) \leq c_2 \mathcal{C}(G) + O(1)$

**Lemma 8.** *Under (A3), there exists $c_2 > 0$ such that*

$$N_\theta^*(G) \leq c_2 \mathcal{C}(G) + O(1).$$

We now explicitly construct a one-layer Transformer that realizes the sparse basis approximations in Assumption (iii).

For each input entity $x$, let
$$S_x := \mathrm{supp}(a_x) = \{j \in [r] : (a_x)_j \neq 0\}.$$

The construction contains one query token representing the input entity $x$ and one learned memory token for each pair $(x, j)$ with $j \in S_x$.

The query token for entity $x$ has query vector $q_x$. Each memory token $(x', j)$ has key vector $k_{x',j}$ and value vector $v_{x',j}$. Choose the keys and queries so that
$$q_x^\top k_{x',j}$$
is large when $x' = x$ and small when $x' \neq x$. For example, assign each entity $x$ an entity code $u_x$ and set

$$q_x = \tau u_x, \qquad k_{x',j} = u_{x'},$$

where $\tau > 0$ controls the attention sharpness and the entity codes are chosen so that $u_x^\top u_x$ is separated from $u_x^\top u_{x'}$ for $x' \neq x$. As $\tau \to \infty$, the attention mass from the query token for $x$ concentrates arbitrarily closely on the memory tokens associated with the same entity $x$.

For the values, let $e_j \in \mathbb{R}^r$ denote the $j$-th standard basis vector. If $s_x := |S_x|$, set

$$v_{x,j} = s_x(a_x)_j e_j \qquad \text{for each } j \in S_x.$$

When attention is uniform over the memory tokens associated with $x$, the attention output is

$$\frac{1}{s_x} \sum_{j \in S_x} s_x(a_x)_j e_j = \sum_{j \in S_x} (a_x)_j e_j = a_x.$$

With sufficiently sharp attention, the retrieved vector can be made arbitrarily close to $a_x$. Thus the query representation after attention is $a_x$ up to an arbitrarily small error.

Finally, choose the shared output matrix to be

$$W_o = B^\top.$$

The resulting logits for input entity $x$ are therefore

$$z_x = W_o a_x = B^\top a_x,$$

up to the arbitrarily small attention approximation error. Applying softmax gives

$$\text{softmax}(z_x) = \text{softmax}(B^\top a_x) = \tilde{p}_x,$$

again up to arbitrarily small error.

By Assumption (iii), $\tilde{p}_x$ is an $\epsilon$-accurate approximation to $p_G(\cdot \mid x)$ for every entity $x$. Therefore this explicit one-layer Transformer achieves $\epsilon$-optimal cross-entropy loss.

It remains to count parameters. The learned memory tokens store the nonzero coefficients of all sparse vectors $\{a_x\}_{x \in \mathcal{E}}$, contributing

$$O\left(\sum_{x \in \mathcal{E}} \|a_x\|_0\right)$$

parameters. The entity-specific query/key codes contribute $O(|\mathcal{E}|)$ parameters up to constant-dimensional coding overhead. The shared output basis contributes

$$O(r|Y|)$$

parameters. Hence the total number of learned parameters is

$$O\left(\sum_{x \in \mathcal{E}} \|a_x\|_0\right) + O(|\mathcal{E}|) + O(r|Y|).$$

Using the sparsity assumption,

$$\sum_{x \in \mathcal{E}} \|a_x\|_0 \leq \sum_{x \in \mathcal{E}} (\alpha H(Y \mid X = x) + \beta) = \alpha C(G) + \beta|\mathcal{E}|.$$

Therefore the constructed Transformer has parameter count

$$O(\alpha C(G) + |\mathcal{E}| + r|Y|).$$

Absorbing constants into the big-$O$ notation gives

$$N_\theta^*(G) \leq \alpha C(G) + O(|\mathcal{E}| + r|Y|).$$

Under the graph regimes considered in this paper, the lower-order terms $|E|$ and $r|Y|$ are either fixed architectural/data-interface costs or are absorbed by the same graph-side scaling assumptions that make total conditional complexity the dominant quantity. Thus the upper bound is

$$N_\theta^*(G) = O(C(G)).$$

**C.5. Conclusion:** $N_\theta^*(G) = \Theta(\mathcal{C}(G))$

Combining Lemmas 7 and 8 yields constants $c_1, c_2 > 0$ such that

$$c_1 \, \mathcal{C}(G) \ \leq \ N_\theta^*(G) \ \leq \ c_2 \, \mathcal{C}(G) + O(1),$$

hence $N_\theta^*(G) = \Theta(\mathcal{C}(G))$.

### C.6. Relating $\mathcal{C}(G)$ to graph search entropy

We briefly justify that $C(G) = \Theta\big(N_e\, \mathbb{E}[L] \cdot (\log \lambda + H^r(G))\big)$.

Let $T = (E_1, R_1, \ldots, E_L, R_L)$, and let the per-step graph search entropy be $h(G) := \log \lambda + H^r(G)$. We define the per-step entropy by MERW is because edges are added with probability proportional to degree: the adjacency matrix has a dominant eigenvalue $\lambda$ that grows with graph size, MERW entropy rate $\log \lambda$ captures the effective branching complexity, and hubs cause slow mixing and large endpoint uncertainty for moderate path lengths.

**Lemma 9.** *Under assumptions (S1) and (S2), for a specific input entity $x$,*

$$H(T \mid X = x) = \Theta\big(h(G)\big).$$

*Proof.* Condition on $L = \ell$. By (S2),

$$H(T \mid X = x, L = \ell) = \ell\, h(G) \pm O(1).$$

Taking expectation over $L$ gives

$$\mathbb{E}[L] \cdot h(G) - C_1 \leq H(T \mid X = x, L) \leq \mathbb{E}[L] \cdot h(G) + C_1$$

for some constant $C_1$. Using the law of total entropy,

$$H(T \mid X = x) \leq H(T, L \mid X = x) = H(L) + H(T \mid X = x, L),$$

and since $L$ is supported on a finite range, $H(L) = O(1)$. The lower bound follows from $H(T \mid X = x) \geq H(T \mid X = x, L)$. $\square$

Then summing over $x \in \mathcal{E}$ gives

$$\mathcal{C}(G) = \sum_{x \in \mathcal{E}} H(Y \mid X = x) = \Theta\big(N_e \cdot (\log \lambda + H^r(G))\big) = \Theta(H(G)),$$

which implies the claimed linear scaling with the proposed graph search entropy up to constants. This completes the proof. $\square$

### C.7. Additional assumptions for tighter bounds.

The constructive upper bound in Theorem 4 yields

$$N_\theta^*(G) \leq O\big(C(G) + |E_{\mathrm{act}}| + r|Y|\big),$$

where $E_{\mathrm{act}}$ denotes the set of entities that appear as query heads, $r$ is the shared basis dimension, and $|Y|$ is the output space size. To obtain a sharper statement of the form $N_\theta^*(G) = \Theta(C(G))$, the additive overhead terms must be controlled. One sufficient condition is a non-degenerate per-head entropy assumption:

$$H(Y \mid X = x) \geq h_{\min} > 0 \qquad \forall x \in E_{\mathrm{act}},$$

which implies $C(G) \geq h_{\min}|E_{\mathrm{act}}|$ and therefore absorbs the entity-indexing cost into $O(C(G))$. Similarly, the shared basis cost can be absorbed by assuming

$$r|Y| = O(C(G)).$$

A tighter characterization also requires the sparse-basis representation to be not only sufficient but near-minimal. In particular, one can assume

$$\sum_{x \in E_{\mathrm{act}}} \|a_x\|_0 = \Theta(C(G)),$$

or, more strongly, $\|a_x\|_0 = \Theta(H(Y \mid X = x))$ for active heads. This rules out cases where the sparse codes are much larger or much smaller than the conditional entropy they represent. For loss-level precision, the approximation condition should be stated directly in KL form:

$$\mathbb{E}_X\big[\mathrm{KL}\big(p_G(\cdot \mid X)\,\|\,\tilde{p}_X\big)\big] \leq \epsilon,$$

so that the constructed predictor achieves at most $\epsilon$ excess cross-entropy, up to the attention-retrieval error.

# D. Data Processing Example

| Type | Example |
|------|---------|
| Original | "triple": {
    "head": "drama film",
    "relation": "/media_common/netflix_genre/titles",
    "tail": "American History X"
} |
| GPT4 generated | "The drama film includes \"American History X\" as one of its Netflix genre titles." |
| Template | "template": "$tail was released as part of the $head genre on Netflix during its period of popularity.",
"sentence": "American History X is featured under the drama film genre on Netflix." |
| Triple-only | (1254, 22, 765) |

*Figure 5.* An example of a triple being processed in three different ways.

# E. Experiment Details

| Model size | hidden size | MLP size | #attention heads | #layers |
|------------|-------------|----------|------------------|---------|
| 0.3M | 128 | 256 | 2 | 2 |
| 0.7M | 128 | 256 | 2 | 4 |
| 1.3M | 256 | 512 | 4 | 2 |
| 2.6M | 256 | 512 | 4 | 4 |
| 5.3M | 256 | 512 | 4 | 8 |
| 10.5M | 512 | 1024 | 8 | 4 |
| 21.0M | 512 | 1024 | 8 | 8 |
| 42.0M | 512 | 1024 | 8 | 16 |
| 83.9M | 1024 | 2048 | 16 | 8 |
| 167.8M | 1024 | 2048 | 16 | 16 |
| 335.6M | 1024 | 2048 | 16 | 32 |
| 671.2M | 2048 | 4096 | 32 | 16 |
| 1342.4M | 2048 | 4096 | 32 | 32 |

*Table 1.* Language model (Llama) size details

| batch size | lr | lr scheduler | warmup ratio | weight decay | max length |
|------------|-----|--------------|--------------|--------------|------------|
| 1024 | 1e-4 | cosine | 0.2 | 0 | 128 |

*Table 2.* Hyperparameter settings for language model pretraining.

|      | $N$              | $N_e$          | $N_r$         | $N_h$      | $\gamma$         |
|------|------------------|----------------|---------------|------------|------------------|
| (a)  | 100k             | 10k            | 100           | 50         | 0.5              |
| (b)  | 10k/20k/.../100k | 10k            | 100           | 50         | 0.5              |
| (c)  | 100k             | 10k            | 100           | 5/10/.../50 | 0.5             |
| (d)  | 100k             | 10k            | 10/20/.../100 | 50         | 0.5              |
| (e)  | 100k             | 10k            | 100           | 50         | 0.1/0.5/.../0.9  |
| (f)  | 10k/20k/.../100k | 1k/2k/.../10k  | 10            | 5          | 0.5              |

*Table 3.* Knowledge graph hyperparameter settings for Figure 3 experiments. We keep $L_{min} = 2$ and $L_{max} = 4$ for all experiments. Here $N$ denotes the number of triples, $N_e$ denotes the number of entities, $N_r$ denotes the number of relations, $N_h$ denotes the number of rules, $\gamma$ denotes the ratio between deductible triples and atomic triples, $L_{min}$ denotes the minimum rule length, and $L_{max}$ denotes the maximum rule length.

## F. Synthetic Knowledge Graph v.s. Real-world Knowledge Graph

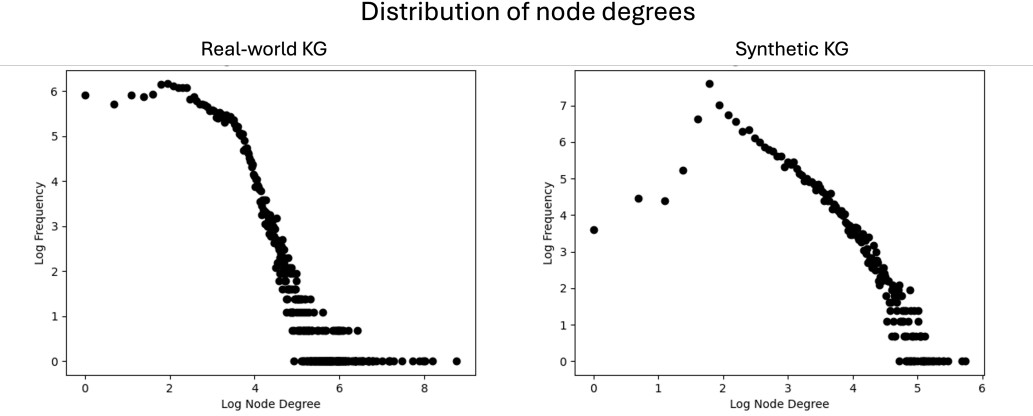

*Figure 6.* Distribution of node degrees of synthetic and real-world knowledge graphs.

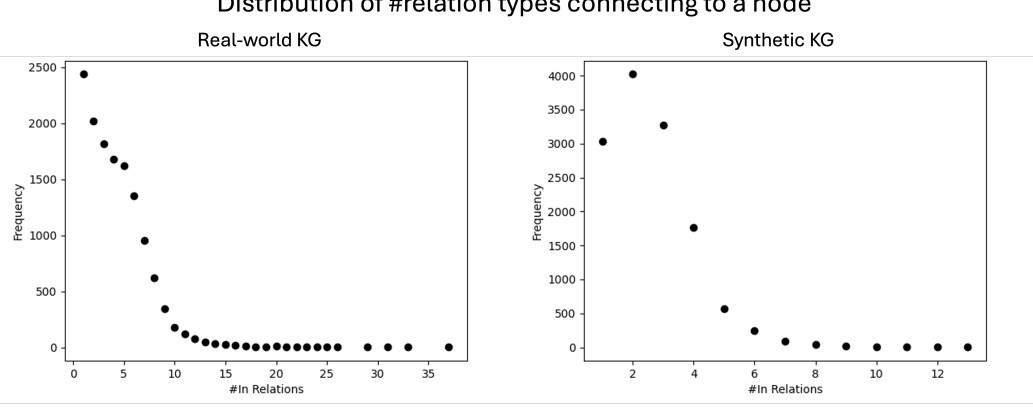

*Figure 7.* Distribution of number of outgoing relations per node of synthetic and real-world knowledge graphs.

## G. Synthetic Knowledge Graph Generation Code

```python
import networkx as nx
import numpy as np
import random
from collections import defaultdict

def add_edge(G, h, t, r):
    num_edges = 0
    if G.has_edge(h, t):
        if r not in G[h][t]['id']:
            G[h][t]['id'].append(r)
            num_edges += 1
        else:
            print('edge already exists')
    else:
        G.add_edge(h, t, id=[r])
        num_edges += 1
    print('add edge: ', (h, r, t), 'num edges: ', num_edges)
    return num_edges

def generate_rules(relations, num_rules, L_min, L_max, weighted=False, temperature=0.25):
    # Generate K acyclic logic rules with varying lengths
    dependency_graph = defaultdict(set)
    rules = []
    weights = []
    if weighted:
        for l in range(L_min, L_max + 1):
            weights.append(np.exp(-temperature*l))
        probs = np.array([w / sum(weights) for w in weights])
    else:
        weights = [1] * (L_max - L_min + 1)

    def has_cycle(start, visited, stack):
        """Detects if adding a new dependency introduces a cycle."""
        if start not in visited:
            visited.add(start)
            stack.add(start)
            print('visited: ', visited)
            print('stack: ', stack)
            for neighbor in dependency_graph[start]:
                if neighbor in stack:
                    return True
                elif has_cycle(neighbor, visited, stack):
                    return True
        if start in stack:
            stack.remove(start)
        return False

    for _ in range(num_rules):
        while True:
            if weighted:
                length = random.choices(range(L_min, L_max + 1), weights=weights)[0]
```

```python
        else:
            length = random.randint(L_min, L_max)
        rule_relations = random.choices(relations, k = length + 1) # the first element is the implied relation
        valid_rule = True
        for i in range(1, len(rule_relations)):
            dependency_graph[rule_relations[0]].add(rule_relations[i])

            # Check for cycles
            if has_cycle(rule_relations[i], set(), set()):
                valid_rule = False
                for j in range(1, i + 1):
                    dependency_graph[rule_relations[0]].remove(rule_relations[j])
                break

        if valid_rule:
            rules.append(tuple(rule_relations))
            break

    print('rules: ', rules)
    return rules

def get_node_types(rules, max_num_relations_per_node=3):
    # map node types to out relations
    node_types = {}
    # map out relations to node types
    r2node_types = defaultdict(list)
    for rule in rules:
        for i in range(len(rule)):
            node_type = len(node_types)
            if i == 0:
                node_types[node_type] = [rule[i], rule[1]]
                r2node_types[rule[i]].append(node_type)
                r2node_types[rule[1]].append(node_type)
            elif i == len(rule) - 1:
                node_types[node_type] = ['-' + rule[i], '-' + rule[0]]
                r2node_types['-' + rule[i]].append(node_type)
                r2node_types['-' + rule[0]].append(node_type)
            else:
                node_types[node_type] = ['-' + rule[i], rule[i+1]]
                r2node_types['-' + rule[i]].append(node_type)
                r2node_types[rule[i+1]].append(node_type)

    print(node_types)
    print(r2node_types)

    for num_rs in range(2, max_num_relations_per_node):
        possible_new_node_types = []
        for r in r2node_types:
            alt_rs = []
            for node_type in r2node_types[r]:
                for _r in node_types[node_type]:
                    if _r != r:
                        alt_rs.append(_r)
            alt_rs = list(set(alt_rs))
```

```
                    for node_type in r2node_types[r]:
                        if len(node_types[node_type]) == num_rs:
                            for _r in alt_rs:
                                if _r not in node_types[node_type]:
                                    possible_new_node_types.append(tuple(sorted([_r] + list(node_types[node_type]))))
                    print(possible_new_node_types)
                    possible_new_node_types += list(set(possible_new_node_types))
                possible_new_node_types = list(set(possible_new_node_types))
                print(possible_new_node_types)

                for rs in possible_new_node_types:
                    new_node_type = len(node_types)
                    node_types[new_node_type] = list(rs)
                    for _r in rs:
                        r2node_types[_r].append(new_node_type)

        return node_types

def get_adj_out_relations(rules):
    adj = defaultdict(list)
    for rule in rules:
        for i in range(len(rule)):
            if i == 0:
                adj[rule[i]].append(rule[1])
                adj[rule[1]].append(rule[i])
            elif i == len(rule) - 1:
                adj['-' + rule[i]].append('-' + rule[0])
                adj['-' + rule[0]].append('-' + rule[i])
            else:
                adj['-' + rule[i]].append(rule[i+1])
                adj[rule[i+1]].append('-' + rule[i])
    return adj

def latent_rule_graph(num_rules=50, L_min=2, L_max=4, n=10000, m=10, n_r=200,
                      num_test=1000, num_train=150000, check_frequency=100,
                      power_law=False, initial_graph=None,
                      length_weighted=False, mcmc=0.2, temperature=0.25,
                      deductible_ratio=0.5):
    # Generate relations and entities
    print("mcmc: ", mcmc)
    relations = ['P' + str(i) for i in range(n_r)]
    all_rules = generate_rules(relations, max(n_r//L_min, num_rules), L_min, L_max)
    r2rules = {}
    for rule in all_rules:
        if rule[0] not in r2rules:
            r2rules[rule[0]] = []
        r2rules[rule[0]].append(rule[1:])
    num_triples = 0
    repeated_entities = defaultdict(list) # map in relation to entities
    child_relations = []
    for rule in all_rules:
        child_relations += rule[1:]
    child_relations = list(set(child_relations))
    child_relations += ['-' + r for r in child_relations]
```

```python
deductible_rules = random.sample(all_rules, num_rules)
if length_weighted:
    weights = [int(100*np.exp(-temperature*len(rule))) for rule in all_rules]
else:
    weights = [1 for _ in all_rules]
repeated_rules = []
for rule, weight in zip(all_rules, weights):
    for _ in range(weight):
        repeated_rules.append(rule)
random.shuffle(repeated_rules)
adj = get_adj_out_relations(repeated_rules)
all_deductibles = {}

if initial_graph is None:
    # Default initial graph
    G = nx.DiGraph()
    node_id = 0
    min_repeated_entities = 0
    while min_repeated_entities < m:
        for rule in all_rules:
            source = 'Q' + str(node_id)
            node_id += 1
            h = source
            for r in rule[1:]:
                t = 'Q' + str(node_id)
                node_id += 1
                num_triples += add_edge(G, h, t, r)
                repeated_entities[r].append(t)
                repeated_entities['-' + r].append(h)
                h = t
            num_triples += add_edge(G, source, t, rule[0])
            repeated_entities[rule[0]].append(t)
            repeated_entities['-' + rule[0]].append(source)

        min_repeated_entities = min([len(set(repeated_entities[r])) for r in child_relations])
else:
    if len(initial_graph) < m or len(initial_graph) > n:
        raise nx.NetworkXError(
            f"Initial graph needs between m={m} and n={n} nodes"
        )
    G = initial_graph.copy()
    node_id = len(G)

if not power_law:
    repeated_entities = {r: list(set(repeated_entities[r])) for r in repeated_entities}

# Start adding the other nodes.
while node_id < n:
    source = 'Q' + str(node_id)
    node_id += 1
    possible_relations = [_r for _r in adj if _r in child_relations]
    if len(possible_relations) == 0:
        print('no adj relations')
        break
```

```python
print('add child edge')
chosen_edges = []
stop = False
for _ in range(m):
    it = 0
    while (r, t) in chosen_edges:
        r = random.choice(possible_relations)
        t = random.choice(repeated_entities[r])
        it += 1
        if it > 100:
            print('failed to find edge')
            stop = True
            break
    if stop or len(possible_relations) == 0:
        break

    possible_relations = [_r for _r in adj[r] if _r in child_relations]
    chosen_edges.append((r, t))
    if r[0] == '-':
        num_triples += add_edge(G, t, source, r[1:])
        repeated_entities[r[1:]].append(source)
    else:
        num_triples += add_edge(G, source, t, r)
        repeated_entities['-' + r].append(source)
    repeated_entities[r].append(t)
    if len(possible_relations) == 0:
        print('no adj relations')
        break

if not power_law:
    repeated_entities = {r: list(set(repeated_entities[r])) for r in repeated_entities}

if node_id % check_frequency == 0 or node_id == n-1:
    # add deductibles
    all_nodes = list(G.nodes)
    random.shuffle(all_nodes)
    for h in all_nodes:
        for rule in deductible_rules:
            head_list = [h]
            r = rule[0]

            for _r in rule[1:]:
                next_head_list = []
                for e_h in head_list:
                    if e_h not in G.nodes:
                        continue
                    for e_t in G[e_h]:
                        if _r in G[e_h][e_t]['id']:
                            if random.random() < mcmc:
                                next_head_list.append(e_t)
                head_list = next_head_list

            for t in head_list:
                if (h, r, t) not in all_deductibles:
```

```python
                    all_deductibles[(h, r, t)] = [rule]
                elif rule not in all_deductibles[(h, r, t)]:
                    all_deductibles[(h, r, t)].append(rule)
                if not G.has_edge(h, t) or r not in G[h][t]['id']:
                    print('add deductible edge')
                    add_edge(G, h, t, r)
                    num_triples += 1
                    repeated_entities[r].append(t)
                    repeated_entities['-' + r].append(h)

atomic_triples = []
deductible_triples = []
for h, t in G.edges:
    for r in G[h][t]['id']:
        if (h, r, t) not in all_deductibles:
            atomic_triples.append((h, r, t))
        else:
            deductible_triples.append((h, r, t))
random.shuffle(atomic_triples)
random.shuffle(deductible_triples)
assert len(atomic_triples) >= int(num_train * (1-deductible_ratio))
assert len(deductible_triples) >= int(num_train * deductible_ratio) + 2 * num_test

remove_triples = []
train_atomic_triples = atomic_triples[:int(num_train * (1-deductible_ratio))]
remove_triples += atomic_triples[int(num_train * (1-deductible_ratio)):]
train_deductible_triples = deductible_triples[:int(num_train * deductible_ratio)]
remove_triples += deductible_triples[int(num_train * deductible_ratio):]

for h, r, t in remove_triples:
    _t = t
    rs = G[h][_t]['id']
    if r in rs:
        if len(rs) == 1:
            G.remove_edge(h, _t)
        else:
            G[h][_t]['id'].remove(r)

train_triples = train_deductible_triples + train_atomic_triples
random.shuffle(train_triples)
print("num train triples: ", len(train_triples))

r2rule = {}
for rule in deductible_rules:
    if rule[0] in r2rule:
        r2rule[rule[0]].append(rule[1:])
    else:
        r2rule[rule[0]] = [rule[1:]]

def check_deductible(triple):
    h, r, t = triple
    alt_ts = []
    for rule in r2rule[r]:
        head_list = [h]
```

```
        for _r in rule:
            next_head_list = []
            for e_h in head_list:
                for e_t in G[e_h]:
                    if _r in G[e_h][e_t]['id']:
                        next_head_list.append(e_t)
            head_list = next_head_list
        alt_ts += head_list
    if t in alt_ts:
        return True
    return False

id_test_triples = []
for i in range(int(num_train * deductible_ratio), len(deductible_triples)):
    if check_deductible(deductible_triples[i]):
        id_test_triples.append(deductible_triples[i])
    if len(id_test_triples) == num_test:
        break

id_test_rules = [all_deductibles[triple] for triple in id_test_triples]
print("num id test triples: ", len(id_test_triples))

rule2triples = defaultdict(list)
for triple in deductible_triples[i+1:]:
    for rule in all_deductibles[triple]:
        rule2triples[rule].append(triple)

# uniformly sample testing triples from each rule
uniform_test_triples = []
for rule in rule2triples:
    triples = []
    for triple in rule2triples[rule]:
        if check_deductible(triple):
            triples.append(triple)

    if len(triples) > num_test//len(rule2triples):
        uniform_test_triples += random.sample(triples, num_test//len(rule2triples))
    else:
        uniform_test_triples += triples

random.shuffle(uniform_test_triples)
uniform_test_rules = [all_deductibles[triple] for triple in uniform_test_triples]
print("num uniform test triples: ", len(uniform_test_triples))

return G, deductible_rules, train_triples, id_test_triples, id_test_rules, uniform_test_triples, uniform_test_rules
```

