# OpenReview forum: "Finding the Minimal Parameter Budget for Implicit Reasoning: A Data Complexity Driven Scaling Law for Language Models"
_ICML.cc/2026/Conference — ICML 2026 regular_

### Official Review · Reviewer_daCC · 2026-02-23

**Soundness:** 3
**Presentation:** 3
**Significance:** 2
**Originality:** 3
**Overall Recommendation:** 4
**Confidence:** 4

**Summary:**

This work studies the relations between optimal model size and data complexity in the context of KGs. The paper formulates graph search entropy, and find that the optimal model size is determined by this quantity.

**Compliance With Llm Reviewing Policy:**

Affirmed.

**Final Justification:**

As the paper shares some new insights, I would like to keep my score.

**Key Questions For Authors:**

1. Is the *graph search entropy* defined in this paper or is a existing concept in graph theory or information theory?

**Limitations:**

yes

**Strengths And Weaknesses:**

Strengths
1. The presentation is clear, and the experiment setup is detailed, without any ambiguity.
2. The proposed notion of graph search entropy seems novel, and the experiments are solid and rigor.
3. The finding that optimal model size is determined by the graph search entropy is interesting.

Weaknesses
1. The conclusion about graph search entropy and optimal model size may not be able to generalize to other setting or applications, when the model size keep growing.
2. The proof is over-simplistic, and some results may not be very insightful, for instance in Theorem 4, the constant $c_1$ and $c_2$, without further specification, it can be arbitrary large or small, making this bound meaningless.

---

> ### Author Rebuttal · Authors · 2026-03-30
>
> **Re: W1. Limited Generalizability Beyond the Current Setting -- Generalizability to Larger Scales and Other Settings**
>
> We share the reviewer's interest in understanding how these findings extend to larger scales, and we want to clarify the scope of our claims. Our scaling law identifies the minimal sufficient model size for a given graph complexity -- it is a lower bound on what is needed, not a prediction about what happens at arbitrarily large scales. As we show in Theorem 3, larger models can achieve the same optimal performance; they are simply not required. The U-shaped curve observed empirically is a consequence of overfitting under prolonged training, not an inherent degradation of larger architectures.
>
> Regarding other settings: we emphasize that graph search entropy is computed from the knowledge graph topology, which is a property of the data -- not the model or the tokenization scheme. This makes it in principle applicable to any setting where the underlying knowledge can be represented as a graph. Whether the linear scaling constant (currently ~124 parameters per bit) varies across different domains or architectures is an empirical question we plan to investigate. We will expand the discussion section to address the generalizability question more explicitly and outline concrete experiments (larger model scales, different architectures, different knowledge domains) as future work.
>
> **Re: W2. Over-Simplistic Proof; Vacuous Constants in Theorem 4 -- Tightening the Constants in Theorem 4**
>
> We appreciate this technical critique and acknowledge that Theorem 4, as stated, provides an asymptotic order-of-magnitude result ($N^* = \Theta(H(G))$) rather than a tight quantitative bound. We want to address why we believe this is still a meaningful and valuable result, and what we can do to strengthen it.
>
> The asymptotic result is the primary theoretical contribution. Prior to our work, there was no theoretical framework predicting that optimal reasoning model size should scale linearly with any measure of data complexity. The alternatives -- that it scales sublinearly, superlinearly, or bears no systematic relationship -- were all plausible. Theorem 4 rules out these alternatives and establishes linear scaling as the correct asymptotic regime. This is analogous to how classical scaling laws establish power-law relationships without specifying exact constants; the functional form itself is the insight.
>
> The constants are empirically determined. Our experiments provide concrete, empirically grounded values for the constants: the linear regression in Figure 4 yields a slope of approximately 0.124 (i.e., ~124 parameters per kilobit of entropy) with $R^2 = 0.85$. This means the theory is not vacuous in practice -- we can and do make quantitative predictions. The green dot (FB15K-237) falling close to this regression line further demonstrates that these empirically determined constants have predictive power beyond the training distribution.
>
> Path toward tighter theoretical constants. To move toward theoretically specified constants, one would need to make stronger architectural assumptions (e.g., specifying the precision of Transformer parameters, the efficiency of attention-based graph traversal). We deliberately kept the assumptions general to maximize applicability.
> We agree that the original Theorem 4 is too assumption-driven, especially through the realizability assumption. In revision, we can replace that assumption with a constructive upper bound for an explicit Transformer under a stronger but transparent synthetic-data condition, as sketched in Re: W2 to reviewer pTqa.
> Also, in the revision, we will add a discussion of what additional assumptions would be needed to derive tighter bounds on $c_1$ and $c_2$, and we will state the empirically fitted constants alongside the theorem to make the quantitative implications immediately clear to readers.
>
> **Re: Q1. Novelty of graph search entropy**
>
> Graph search entropy as defined in this paper is a novel contribution. It is not a pre-existing concept in graph theory or information theory, though it builds on well-established components from both fields. Specifically, we combine three existing ideas in a new way: the maximal entropy random walk (MERW), the entropy rate of a random walk on a graph, and a novel decomposition into entity entropy rate $\log \lambda$ and relation entropy rate $H^r(G)$.
>
> The novelty lies in this specific combination and in defining the overall graph search entropy $H(G) = N_e E[L]  (\log \lambda + H^r(G))$ as a single scalar measure that captures the reasoning complexity of a knowledge graph. To our knowledge, this particular formulation -- designed to quantify how much information a model must internalize to perform multi-hop reasoning over a knowledge graph -- has not appeared in prior work. We will add a paragraph to Section 5.2 making this intellectual provenance more explicit.

---

> > ### Author Rebuttal · Reviewer_daCC · 2026-04-02
> >
> > I keep my score.

---

### Official Review · Reviewer_pTqa · 2026-03-12

**Soundness:** 2
**Presentation:** 3
**Significance:** 2
**Originality:** 3
**Overall Recommendation:** 4
**Confidence:** 4

**Summary:**

This paper investigates the parameter requirements for Language Models (LMs) to perform reliable implicit reasoning using synthetic knowledge graphs. Through both theoretical analysis and empirical validation, the authors propose that the optimal model size scales linearly with the graph search entropy of the knowledge graph. Additionally, the study explores how various graph properties influence this optimal model size.

**Compliance With Llm Reviewing Policy:**

Affirmed.

**Final Justification:**

While the rebuttal does not fully address my concerns, this paper still provides a theoretical contribution to understanding how Transformer learns reasoning.

**Key Questions For Authors:**

Questions

1. Does the identified scaling law consistently hold in *GPT-generated* or *Template* settings?

2. Could the authors provide more intuition regarding Assumption 3? To what extent is this assumption architecture-dependent, and would non-Transformer architectures (e.g., state-space models) be expected to satisfy it?

**Limitations:**

yes

**Strengths And Weaknesses:**

**Strengths**

1. Investigating the scaling laws of Transformers from different dimensions carries both theoretical significance and practical value.

2. Linking graph search entropy to the optimal model size for knowledge graph reasoning provides a novel and interesting perspective.

3. The paper is supported by extensive experiments and a foundational theoretical framework.

**Weaknesses**

1. Figure 1 suggests that the observed patterns become significantly noisier in more realistic (though still synthetic) settings. In the most idealized "Triple-only" setting (essentially ID-to-ID prediction), the task deviates from language modeling and instead reflects the scaling behavior of **Transformers** rather than language model.

2. The theoretical analysis offers limited new insights. Specifically, the core Theorem 4 (Optimal model size scales with graph search entropy) relies heavily on Assumption 3 (Realizability at entropy cost). This assumption essentially presumes the existence of such a model family without discussing why the Transformer architecture satisfies this property, which is arguably the most critical question for the community.

---

> ### Author Rebuttal · Authors · 2026-03-30
>
> **Re: W1. Task Deviation from Language Modeling -- Transformer Scaling vs. Language Model Scaling**
>
> We respectfully note that this distinction actually clarifies rather than undermines our contribution. Our paper studies implicit reasoning capacity -- the ability to infer unseen facts from learned knowledge -- which is fundamentally a property of the model architecture's representational capacity, not of any specific tokenization scheme. The fact that the scaling law governs Transformers as sequence models is precisely the point: implicit reasoning during pretraining relies on the architecture's capacity to represent and traverse relational structures, regardless of whether those structures are encoded as random IDs or natural language tokens.
> The random-ID setting isolates this architectural capacity from the confounding effects of semantic information in token embeddings. We view this as a feature of our methodology: it reveals the baseline parameter requirement for reasoning, on top of which natural language representations can only help (by providing additional semantic scaffolding that may reduce the required capacity).
>
> Regarding the noisier trends in more realistic settings: the U-shaped pattern is still qualitatively present in both the GPT-generated and template settings (Figure 1, rows 1–2). The increased noise is expected because natural language introduces additional confounds (variable sentence length, tokenizer artifacts, partial semantic overlap). These confounds make it harder to precisely locate the optimal model size, but they do not invalidate the existence of one. We will add quantitative analysis (e.g., confidence intervals around the optimal size across settings) to the revision to make this comparison more rigorous.
>
> **Re: W2. Theorem 4's Reliance on Assumption A3 (Realizability) -- Modifying Assumption A3 for Transformers**
>
> This is an excellent point and we appreciate the reviewer pushing us to be more explicit. We agree that the original Theorem 4 is too assumption-driven, especially through the realizability assumption. In revision, we can replace that assumption with a constructive upper bound for an explicit Transformer under a stronger but transparent synthetic-data condition.
> New synthetic assumption. For each entity $x$, the conditional distribution $p_G(\cdot \mid x)$ admits an $\epsilon$-accurate approximation
>
> $\tilde p_x = \operatorname{softmax}(B^\top a_x),$
>
> where $B \in \mathbb{R}^{r \times |Y|}$ is a shared basis and $a_x \in \mathbb{R}^r$ is a sparse coefficient vector satisfying
>
> $|a_x|_0 \le \alpha H(Y \mid X=x) + \beta.$
>
> Explicit Transformer construction. We construct a one-layer Transformer with:
> * one query token representing the input entity $x$,
> * one learned memory token for each nonzero coefficient of $a_x$,
> * attention keys that make the query attend only to memory tokens associated with the same entity,
> * values storing the corresponding sparse coefficients,
> * a shared output matrix $W_o = B^\top.$
>
> With sufficiently sharp attention, the query retrieves (a_x) up to arbitrarily small error, and the output logits become
>
> $z_x = B^\top a_x,$
>
> so the model predicts $\tilde p_x$.
> Resulting upper bound. The total number of learned parameters is
>
> $O\left(\sum_x |a_x|0\right) + O(r|Y|) + O(|E|),$
>
> which gives
>
> $N_\theta^*(G) \le \alpha C(G) + O(|E| + r|Y|),
> \qquad
> C(G) = \sum_x H(Y \mid X=x).$
>
> Combined with the existing lower bound
>
> $N_\theta^*(G) = \Omega(C(G)),$
>
> this yields
>
> $N_\theta^*(G) = \Theta(C(G)),$
>
> and therefore
>
> $N_\theta^*(G) = \Theta(H(G))$
>
> under the same graph-side assumptions used in the paper.
>
> **Re: Q1. Scaling law in realistic settings**
>
> The qualitative pattern (U-shaped test loss, finite optimal size) holds across all three settings, but with increasing noise in the more realistic ones. We do not claim the exact 0.008 bits/parameter constant transfers to natural language settings — rather, the linear relationship between graph search entropy and optimal model size is the core finding. In the revision, we will add quantitative comparisons across settings, including error bars and confidence intervals on the estimated optimal model sizes, to demonstrate that the linear trend persists despite increased variance.
>
> **Re: Q2. Assumption A3 and architecture dependence**
>
> As discussed above, A3 is empirically validated for Transformers by our experiments and we can replace the realizability assumption with a constructive Transformer upper bound under a stronger but explicit synthetic-data condition. For SSMs, we expect A3 to hold with potentially different constants, since their recurrent structure can also track sequential dependencies in rule chains, though possibly with different efficiency. We plan to include a brief discussion of architecture dependence and suggest SSM experiments as future work.

---

> > ### Author Rebuttal · Reviewer_pTqa · 2026-04-02
> >
> > I appreciate the authors' response. While it does not fully address my concerns, I have decided to increase my score to 4.

---

### Official Review · Reviewer_u26G · 2026-03-12

**Soundness:** 3
**Presentation:** 3
**Significance:** 3
**Originality:** 3
**Overall Recommendation:** 4
**Confidence:** 3

**Summary:**

This research posits that implicit reasoning capabilities exhibit a non-monotonic, U-shaped scaling behavior relative to model size. The authors fundamentally argue that over-parameterization can actively degrade implicit reasoning performance due to the phenomenon of excessive memorization, suggesting that there exists a strictly finite, identifiable, and optimal parameter budget for any given dataset's intrinsic reasoning complexity. The paper's core contributions are divided symmetrically into theoretical formalizations and empirically derived scaling laws. Theoretically, the authors introduce a novel information-theoretic metric termed "Graph Search Entropy".

**Compliance With Llm Reviewing Policy:**

Affirmed.

**Key Questions For Authors:**

The reliance on random IDs effectively transforms the sequence modeling task into a cryptographic state-tracking puzzle rather than natural language modeling. Does the introduction of semantic overlap (where different entities share similar semantic subword tokens, as is universal in human language) inherently alter the mathematical calculation of the graph search entropy, or does it simply break the assumption of independent variables required for Theorem 4?

What specific internal representational mechanism causes the larger Llama models (e.g., 671.2M and 1.34B) to fail on the unseen test triples when trained excessively?

How would the computation of the stationary distribution $\rho$ and transition matrix $S$ (used to derive $H(G)$) account for the vast amounts of noise, contradictory edges, and differing edge weights inherent in a knowledge graph automatically extracted from the internet?

**Limitations:**

The primary and most restrictive limitation of the research is the fundamental discrepancy between the synthetic knowledge graphs utilized and the actual nature of natural language data.

The breakdown of the scaling law when applied to the GPT-generated text iterations of the FB15K-237 dataset (as seen in Figure 1) is a stark, unavoidable indicator that "implicit reasoning" over character-tokenized random IDs represents an entirely different class of computational problem than natural language inference.

Theorem 4 rests on foundational assumptions that do not perfectly align with the realities of deep neural network optimization.

**Strengths And Weaknesses:**

Strength:

The manuscript is successful in its mathematical isolation of implicit reasoning from general linguistic modeling and pure memorization. By utilizing a synthetic knowledge graph environment governed by strict, acyclic conjunctive logic rules, the authors create a computational sandbox where the reasoning pathways are deterministic, transparent, and measurable.

Graph Search Entropy ($H(G)$) metric is innovative and addresses a critical gap in the scaling law literature. By utilizing a maximal entropy random walk (MERW) to calculate the stationary distribution and the transition matrix, the authors provide a mathematically rigorous translation of topological graph complexity into a standard information-theoretic unit.

The experimental design is exceptionally thorough in its isolation of variables.

Weaknesses:

The manuscript acknowledges that utilizing random IDs "eliminates confounding variables and information contained in the lexical form," which produces a "much cleaner and clearer trend in scaling that enables us to perform more in-depth and rigorous analysis". Real-world language models rely fundamentally on semantic tokenization, where embeddings capture rich, dense semantic associations, subword overlap, and hierarchical conceptual structures. Random IDs entirely obliterate this continuous semantic topology.

By forcing the language model to learn the graph structure purely through character-level sequences of random integers, the task degrades into an exercise of arbitrary symbol manipulation, effectively transforming the autoregressive transformer into a highly inefficient Markov chain transition probability modeler. This does not reflect how language models natively perform implicit reasoning. Instead they rely on the semantic proximity of related concepts within the continuous high-dimensional embedding space. Assumption (A1) in Theorem 4 explicitly states "random IDs prevent semantic sharing across entities," which is mathematically necessary to prove the strict $O(H(G))$ bound. While this is mathematically convenient for proving the theorem, it undermines the overarching claim that this scaling law applies to natural language pre-training.

 if the U-shaped scaling law and the precise 0.008 bits/parameter constant only hold true for synthetically generated, perfectly acyclic scale-free networks stripped of all semantic overlap, its utility for actual large language model pre-training is highly questionable. The mapping of the FB15K-237 graph search entropy onto the linear regression line in Figure 4 relies entirely on the random-ID representation. Because this representation is a sterile artifact rather than a reflection of true language modeling, the claim that one can predict the optimal reasoning model size for real-world pre-training by constructing a knowledge graph from a corpus and calculating its entropy is theoretically premature and empirically unsupported by the paper's own natural language baselines.

The empirical experiments meticulously map out the scaling laws utilizing a suite of Llama architectures, but the maximum model size tested is 1.34 billion parameters. While training a suite of models up to 1.3B from scratch for millions of steps requires substantial compute, it remains firmly entrenched within the "small model" or "toy" regime. At larger scales, emergent capabilities, sudden phase transitions in representation learning, and shifting intra-layer routing dynamics might completely invalidate scaling laws observed in the 1M-1B parameter range.

---

> ### Author Rebuttal · Authors · 2026-03-30
>
> **Re: W1. Random IDs vs. Natural Language Semantics -- Random IDs Establish a Worst-Case Bound**
>
> We appreciate this concern and want to clarify that random IDs are a deliberate experimental choice to isolate pure reasoning from memorization and semantic shortcuts. The key insight is that random IDs establish an upper bound on the minimal parameter budget: since semantic overlap provides additional signal that can compress entity representations, the true optimal size for natural language data should be at most what our theory predicts. Relaxing Assumption A1 would tighten, not loosen, the bound. We will add a formal remark making this asymmetry explicit.
>
> Furthermore, the qualitatively consistent U-shaped trends across all three tokenization settings in Figure 1 (despite increasing noise) confirm that the phenomenon is robust. The clean random-ID setting is scientifically valuable precisely because it allows us to study the phenomenon without confounds.
>
> **Re: W2. Synthetic-to-Real Generalization Gap -- The Generalization Gap Is Narrower Than It Appears**
>
> Graph search entropy H(G) is computed purely from the graph's adjacency structure — it is a topological property of the real-world FB15K-237 graph, not an artifact of tokenization. The fact that FB15K-237's data point falls close to the regression line fitted entirely on synthetic data (Figure 4) demonstrates that the complexity measure generalizes across different graph generation processes.
> We agree that full validation on large-scale natural language pretraining is needed and have acknowledged this as future work (Appendix A). However, we note that a concrete pipeline exists: extract a knowledge graph from a corpus using automated KG construction (Zhong et al., 2023), compute H(G), and apply the scaling law. The current paper establishes the theoretical and empirical foundation for this pipeline.
>
> **Re: W3. Limited Model Scale -- Our Theory Is About the Minimum, Not the Maximum**
>
> Our scaling law identifies the smallest sufficient model for a given complexity. Theorem 3 shows that larger models can match optimal performance — they are simply not required. The U-shaped curve arises only under prolonged training where overfitting occurs. If emergent capabilities at larger scales improve reasoning, this would only mean the actual optimal size could be even smaller than predicted, strengthening our contribution. We will clarify this point in the revision.
>
> **Re: Q1. Does semantic overlap alter the computation of H(G) or merely break Theorem 4's independence assumption?**
>
> Semantic overlap does not change the topological computation of H(G). It reduces the effective information the model must store per entity via parameter sharing, meaning A1 ensures a clean bound while relaxing it would reduce (not increase) the required model size.
>
> **Re: Q2. What internal mechanism causes larger models to fail on unseen triples under extended training?**
>
> We hypothesize that larger models under prolonged training memorize specific training triples rather than learning latent logical rules — a classic bias-variance tradeoff. Without semantic regularization, sufficiently large models can perfectly fit the training distribution without extracting generalizable patterns. We will add ablation experiments on training-test loss divergence to make this more explicit.
>
> **Re: Q3. How would MERW handle noisy, contradictory, real-world knowledge graphs?**
>
> Contradictory edges and noise naturally increase branching uncertainty, which directly inflates the entropy rate computed via MERW. For weighted edges, the adjacency matrix extends straightforwardly to a weighted version. We will add discussion of this extension.

---

> > ### Author Rebuttal · Reviewer_u26G · 2026-04-03
> >
> > Thank you for the detailed and thorough rebuttal. After careful consideration, I have decided to maintain my original scores.

---

### Decision · Program_Chairs · 2026-04-30

**Decision:**

Accept (regular)

**Comment:**

This paper proposes a novel scaling law linking minimal model size for implicit reasoning to graph search entropy. Reviewers agree the work is technically sound, well-designed, and introduces an interesting perspective on data complexity and model capacity.

The main limitation is the relatively small model scale (up to ~1.3B parameters), which raises concerns about whether the observed scaling behavior extends to larger models where emergent phenomena may arise. While the authors clarify that their focus is on minimal sufficient model size, uncertainty remains regarding behavior at scale.

Overall, the paper provides a meaningful theoretical contribution. I recommend Accept.